# Chinese Students’ Health Literacy Level and Its Associated Factors: A Meta-Analysis

**DOI:** 10.3390/ijerph18010204

**Published:** 2020-12-29

**Authors:** Ying Mao, Tao Xie, Ning Zhang

**Affiliations:** School of Public Policy and Administration, Xi’an Jiaotong University, Xi’an 710049, China; xietao2014077049@stu.xjtu.edu.cn (T.X.); zhangningati@stu.xjtu.edu.cn (N.Z.)

**Keywords:** health literacy, associated factors, Chinese students, meta-analysis

## Abstract

Health Literacy (HL) is an important determinant of health. Many scholars have conducted a large number of studies on the level of Chinese students’ HL and its associated factors. However, previous studies on HL level and the factors that influence it have been contradictory. Therefore, this systematic review and meta-analysis was conducted to estimate the level of Chinese students’ HL and its three dimensions (knowledge, behavior and skills) and to identify factors associated with HL in Chinese students. Two investigators independently searched literature, selected research and extracted data through comprehensively searching of four international electronic databases and three Chinese electronic database to identify all relevant observational studies on affecting factors for HL in Chinese students published in English and Chinese from database January, 2010 to September, 2020. In total, 61 articles were extracted in the study. The results showed that the level rates of HL and its three dimensions were 26%, 35%, 26%, 51%, respectively. For Chinese students, the significant factors were urban residents, senior class students, well performance at school, the Han nationality, focus on health knowledge, less exposure to video games, highly educated parents, income of one-child families, receiving health education, having medical background. This study provides some inspirations for improving the level of Chinese students’ HL and their health. First, the findings may help Chinese policy makers understand the overall HL of Chinese students and their levels across three dimensions (knowledge, behavior and skills). Second, protective factors for Chinese students’ HL were found in this research, which will help to improve the level of Chinese students’ HL, stimulate students’ awareness of prevention, and lay the foundation for a healthy China.

## 1. Introduction

At present, experts and scholars have gradually reached a consensus on the definition of Health Literacy (HL). HL refers to the ability of individuals to obtain, understand and process basic health information and services, and use such information and services to make decisions that are conducive to improving and maintaining their own health [1,2]. Chinese scholars have also introduced this concept, which has been recognized by the health authorities [3].

HL is an important determinant of health and a comprehensive reflection of the level of economic and social development. It is restricted by political, economic, cultural, educational and other factors [4]. The study of the World Health Organization (WHO) [4,5] shows that HL is an effective indicator to predict the health status of the population, which is highly correlated with morbidity, mortality, health level, average life expectancy and quality of life. Therefore, the WHO has promoted the promotion of public HL level as an important strategy and measure to improve public health worldwide. According to the National Health and Family Planning Commission of China, the immediate aim of improving public HL is to prevent, reduce or delay the onset of illness, so that the patients are enabled to better manage the diseases they are suffering from, prevent the deterioration of the diseases, and reduce the number of illness, disability and death [6].

The student period is not only an important stage to acquire knowledge, but also a key stage to develop behavioral habits. Moreover, primary and middle school students are in the critical period of physical and mental development with strong plasticity, which is the stage with the best effect and best opportunity to carry out HL intervention [7]. The improvement of primary and secondary school students’ HL not only has a profound impact on their health throughout their lives, but also has a positive influence on their families and even the whole society [8].

Therefore, from the perspective of health promotion, it is important to recognize educated children and teenagers as the target group for HL research and interventions: childhood and adolescence are seen as essential life stages for healthy development and well-being of individuals throughout adulthood [9], because key pathways to future health have evolved over the course of life and become apparent in children and youth period [10]. It has been clearly stated that failure to cultivate young people access to health knowledge and their capability of promoting health, not only could increase potential personal and social risks, but also might cause worse health outcomes and higher costs [11,12,13].

On 24 April 2020, the National HL Monitoring Department of the National Health Commission released the 2019 national HL monitoring results, indicating that the HL level of Chinese residents rose to 19.17%. Compared to 2018, in 2019 the basic knowledge and ideological level of Chinese residents was 34.31%, which saw an increment of 3.79 percentage points; The percentage of the level of healthy lifestyle and behavioral literacy increased by 2.44 percentage points at 19.48%; The level of basic skills and literacy was 21.43%, which experienced a growth of 2.75 percentage points [14]. The above three aspects of knowledge, behavior and skills are all improved respectively.

However, the HL survey mentioned above mainly focuses on people aged 15 to 69. Also, few studies on the HL of primary and middle school students are conducted, and college students are not included in the monitored population.

In order to improve the level of Chinese students’ HL, the identification of the protective factors associated with it is extremely critical. So far, many scholars have conducted a large number of studies on the level of Chinese students’ HL and its associated factors. Nevertheless, research on HL levels and the factors that influence them has been contradictory. For example, some studies found some differences in the level of Chinese students’ HL. Other studies [15,16,17,18] showed that no relationship was found between HL level and students’ gender, whereas other scholars hold the opposite view [19,20,21]. Furthermore, conflicting results about the relationship between HL and students’ residence were found. Various studies suggested that the HL level of urban students is higher than that of rural students [22,23,24,25,26,27,28,29,30,31,32]. On the contrary, according to other studies, rural students have a higher level of HL [29]. There is also debate about grade level. Some studies argued that students in higher grades have a higher level of HL [23,28,33,34,35,36,37,38,39,40], while other researchers suggested students in lower grades have a higher level of HL [24,41].

As shown above, the factors affecting the HL of Chinese students have not been concluded yet, and there are still many controversies about the correlation between various factors and HL in existing studies. To our knowledge, no meta-analysis has been conducted to clarify the controversial relationship between HL and its associated factors. Thus, the meta-analysis to study the potential relationship between these variables and HL can provide a valuable reference for the study of whether they are relevant and whether they should be included in the associated factors of HL. The study from this perspective provides scientific basis for targeted intervention of Chinese students’ HL, so as to improve the level of Chinese students’ HL.

Considering the above factors, the research aims of this study are as follows: (a) to estimate the level of Chinese students’ HL and its three dimensions (knowledge, behavior and skills) through meta-analysis; (b) A meta-analysis was to be conducted on the correlation between Chinese students’ HL and various factors.

## 2. Methods

### 2.1. Search Strategies

Based on the guidelines for systematic evaluation and meta-analysis (known as PRISMA; Appendix A), the authors conducted a comprehensive search of four international electronic databases (PubMed, Embase, Web of Science, Cochrane Library) and three Chinese electronic database (CNKI, Wan Fang, CQVIP) to identify all relevant observational studies on affecting factors for HL in students in China published in English and Chinese from database 1st January 2010 to 14th September 2020. The same terminology is used each time the database is searched: (1) “HL” or “literacy” or “health”; (2) “China” or “Chinese”; (3) “Student” or “School Enrollment” or “Enrollment, School” or “Enrollments, School” or “School Enrollments; (4) “pupil” or “elementary student” or “middle school student” or “junior; senior” or “high school student” or “undergraduate” or “academician” or “university man; (5) “relative risk” or “factors”. These search themes were combined using the Boolean operator “and” in without restrictions. In addition, a list of references and recent reviews of the retrieval papers are also reviewed.

### 2.2. Selection Criterion

Studies that met all of the following criterion were included: (1) full-text is available; (2) conducted in China; (3) participants were students; (4) samples were from the general population; (5) published in English language in international electronic databases; (6) published between 2010 and 2020; (7) the study reported the level of HL and relevant affecting factors; (8) must utilize an authoritative definition of HL; (9) OR (odds ratio) and 95%CI can be extracted for effective research factors.

Studies that met all of the following criterion were excluded: (1) only titles and abstracts are available; (2) not conducted in China; (3) not have definitions about HL; (4) samples are not clear; (5) retrospective studies; (6) unclear statistical methods; (7) eHealth Literacy (E-HL). Two investigators independently screened the titles and abstracts of articles retrieved from the literature search, and evaluated the eligibility of all studies on the basis of the predetermined selection criteria. Disagreements were resolved through discussions between investigators or in consultation with the third one.

### 2.3. Data Extraction

Two investigators (Xie Tao and Zhang Ning) independently extracted relevant data from each eligible study. A preconceived and standardized form was used to extract data. Information on study characteristics extracted included: first author’s name, publication year, investigation year, sample size, source of questionnaire, questionnaire recovery, the overall level of HL, the HL level of basic knowledge, the HL level of healthy lifestyle and behaviors, the HL level of health skills and the number of participants. Any disagreements between authors were resolved by consensus and discussion.

### 2.4. Quality Assessment of Included Studies

This study used the Agency for Healthcare Research and Quality (AHRQ) [42] to independently evaluate the included studies by two investigators (Xie Tao and Zhang Ning). Each item was assessed a score of 1 (yes) or 0 (no or unclear), and summed scores for all items to get an overall quality score from 0 to 11. According to the overall scores, study quality was classified as follows: low (0–3); moderate (4–7); high (8–11). Disagreements resolved by consensus and discussion.

### 2.5. Factors of Induction

Through discussion, we divide many factors into four dimensions: Individual characteristics (IC), Behavioral habits (BH), Family environment (FE), School education (SE). IC included gender, locations, grade, academic performance, and race, among IC, race only involved college students. BH included focus on health knowledge and exposure to video games, among which attention to health information only involved college students. FE included father’s education level, mother’s education level, single-child and family income, among which the family income factor only involved college students. SE included health education courses, majors and school types, among which only majors and school types did not involve primary and middle school students.

### 2.6. Statistical Analysis

Estimates for specific studies are summarized to obtain an overall summary value of level of HL and its three dimensions (knowledge, behavior and skills). OR and its 95% CI were combined to evaluate the associated factors of HL. All of the above data merging was done by using the Stata version 15.0 for Mac (StataCorp, College Station, TX, USA). The data analysis results were presented in the form of forest plots to determine whether there was a statistical association. The heterogeneity of the pooled results was quantified evaluated using the Cochrane’s Q test and Higgins’ I-squared. The *p*-value < 0.1 or I^2^ ≥ 25% was considered have heterogeneity and pooled the overall result using the random effects model. Otherwise, the influence of heterogeneity would be ignored and the fixed effects model was employed.

In addition, the researchers examined the impact of individual studies on overall risk estimates by removing each study in each round to test the robustness. The potential publication bias was assessed by Egger test. The *p* value of Egger test less than 0.05 was considered significant publication bias.

## 3. Results

### 3.1. Characteristics of the Study Samples

Through the systematic search of literature, 1658 unique records were identified for screening, and 1326 records were left after the removal of repetitive literatures. By scanning the titles and abstracts, the authors excluded 1144 unrelated entries. The researchers evaluated the full text of the remaining 183 papers to determine eligibility, 98 of which were excluded. 85 full-text articles were included, of which only 61 studies could be pooled for analysis. The study selection process was documented in the flow chart in Figure 1.

The characteristics and quality of the included studies are shown in Table 1. There were 21 researches of primary and middle school students, 40 of which were college students and 16 of college students were medical students. Most of the publications are from recent years. One study did not provide the overall level rate of HL, and seven did not provide the level rate of basic knowledge and ideas of HL, level rate of healthy lifestyle and behavior, and level rate of health skills. The mean values of the above level rates were 25.68%, 34.65%, 25.74%, and 50.94%, respectively. 5 studies did not provide effective recovery rates of questionnaires, the average of which was 96.04%. Respondents who correctly answered 80% or more of the health literacy questions were considered to have HL. In terms of knowledge, behavior, and skills correctly answering questions of 80% or more is regarded as having health literacy of this dimension expect one article [43] which was judged to have the HL score not ≥80%. Except for the four studies [22,43,44,45], most of the studies clearly mentioned that the questionnaire was designed based on the HL questionnaire of Chinese citizens and its various derivative versions. The total number of samples was 154,113. According to AHRQ’s evaluation of research quality, the quality score of each study was presented in Table 1, 13 were of high quality, 48 were moderate, and there were no articles with low quality rating.

### 3.2. Meta-Analysis on Main Factors and Level of HL

Meta-analysis results were divided into two parts: one was the combination of HL level (See the annex for the forest map), and the other was the combination of associated factors of HL. The overall HL level of Chinese students was 26% (95% CI, 21–30%), and the basic knowledge and concept of HL, healthy lifestyle and behavior level, and health skills level were 35% (95% CI, 29–40%), 26% (95% CI, 19–33%), 51% (95% CI, 45–57%), respectively. All the above results were obtained by random effects model (heterogeneity test: I^2^ > 90%, *p* < 0.001). In Egger’s test, the authors found publication bias for overall HL (Egger’s test *p* = 0.025) and the basic knowledge and concept of HL (Egger’s test *p* < 0.001).

It could be seen from Figure 2 (ES was effect size, the equivalent of OR in this study) that the results of other related factors excepted race (I^2^ = 27.4%, *p* = 0.252) had obvious heterogeneity (I^2^ > 50%, *p* < 0.1). Therefore, after the sensitivity analysis was continued, some studies were eliminated and meta-analysis was conducted again. The results were shown in Figure 2. It could be seen that although gender and locations had slight heterogeneity, which was within the acceptable range. Therefore, random effects were selected to conduct a meta-analysis of IC, and it was concluded that gender (Z = 6.6, *p* < 0.05), locations (Z = 6.78, *p* < 0.05), grade (Z = 10.97, *p* < 0.05), academic performance (Z = 9.59, *p* < 0.05), Race (Z = 2.89, *p* < 0.05) were associated with HL. In general, the probability of female, urban residents, senior students, good academic performance and Han students having HL was1.45 times (95% CI, 1.30–1.62), 1.7 times (95% CI, 1.46–1.98), 2.34 times (95% CI, 2.01–2.73), 1.54 (95% CI, 1.41–1.68) times and 1.55(95% CI, 1.15–2.08) times of male, rural students, lower grade students, poor academic performance and ethnic minorities, respectively. 0.135). In addition, the results shown publication bias for locations (Egger’s test *p* = 0.007) and academic performance (Egger’s test *p* = 0.012).

Figure 3 showed obvious heterogeneity in the results of all studies on BH. After sensitivity analysis, relevant studies were excluded, and meta-analysis was performed again. Figure 2 found after conducting meta-analysis of the health information attention heterogeneity decreased obviously (I^2^ = 25.40%, *p* = 0.252), and thus fixed effects were selected for meta-analysis. Exposure to video games still had high heterogeneity and random effects were selected for meta-analysis. It was concluded that students who pay attention to health information (OR2.78, 95% CI, 1.96–3.95) and play online games for less than 5 h (OR 0.24, 95% CI, 0.08–0.71) have HL. In addition, the authors found publication bias for health information attention (Egger’s test *p* = 0.023).

Meta-analysis results (Figure 4) showed that the heterogeneity of the four associated factors of FE was large. After the exclusion of relevant studies by sensitivity analysis, the heterogeneity of father’s education level and mother’s education level decreased (I^2^ = 25.10%, *p* = 0.171; I^2^ = 12.00%, *p* = 0.325), and was significantly associated with HL (Z = 11.06, *p* < 0.001; Z = 9.44, *p* < 0.001). Therefore, fixed effect was selected for analysis, and it was found that higher educational level of fathers (OR 1.86, 95% CI, 1.7–2.04) and mothers (OR 1.76, 95% CI, 1.58–1.96) was associated with higher probability of HL of their children. Although the heterogeneity of single-child and family income also decreases, it still has certain heterogeneity. As a result, random effect was selected for analysis. The researchers found these two factors were significantly correlated with HL (Z = 3.28, *p* < 0.05; Z = 5.44, *p* < 0.001), and only children (OR 2.23, 95% CI, 2.38–3.62) and higher family income level (5.06, 95% CI, 2.82–9.08) are protective factors of HL. In addition, the results suggested publication bias for parents’ education level (farther: Egger’s test *p* = 0.001; mother: Egger’s test *p* = 0.002) and single-child (Egger’s test *p* = 0.043).

The results of SE (Figure 5) suggested that all of them had high heterogeneity. After sensitivity analysis, meta-analysis again found that only school type still had high heterogeneity, and therefore the random effect model was selected for analysis. Due to low heterogeneity of health education courses and major (I^2^ = 4.30%, *p* = 0.394; I^2^ = 25.10%, *p* = 0.229) fixed effect model was used for analysis. The results showed that school education was significantly correlated with HL (Z = 6.97, *p* < 0.05; Z = 14.11, *p* < 0.001; Z = 3.87, *p* < 0.001), students who study health education courses (OR 1.77, 95% CI, 1.51–2.07), medical majors (OR 3.75, 95% CI, 3.21–4.39) and medical schools (OR 0.35, 95% CI, 0.2–0.59) have a high probability of having HL. Moreover, publication bias for health education courses were found (Egger’s test *p* = 0.045).

## 4. Discussion

At present, there has been no meta-analysis on the associated factors of Chinese students’ HL. This comprehensive research of relevant articles in Both Chinese and English provided strong evidence for the influential factors of Chinese students’ HL. The purposes of this study were to calculate the effect size of the associated factors of HL and to examine the level of HL. The results showed that IC, BH, FE and SE were highly correlated with HL level. The overall level of HL of Chinese students was higher than that of residents, including knowledge, behavior and skills.

### 4.1. Impacts of IC on HL

The analysis results showed that IC had the following five protective factors: Female [19,20,21,27,34,37,41,43,59,78], Urban [22,23,24,25,26,27,28,29,31,32,64], Senior [23,28,33,34,35,36,37,38,39,40,68], Good academic performance [22,26,41,45,51,55,64,65], Han ethnic [38,66,72], which was consistent with other relevant research results in China. The possible reasons for why women’s HL level is higher than that of men might be that women are more careful, they pay more attention to the details of life and prefer to clean (It might not be very persuasive since some of the reasons are subjective. It is suggested to list some objective facts rather than opinions) and hygiene and would like to open to health-related knowledge, and thus they are more likely to cultivate healthy lifestyle and good personal hygiene habits. At the same time, under the influence of the education thoughts of men and women, Chinese parents pay more attention to girls and pay more attention to the care of their lives, which will also lead to the cultivation of good living habits.

City HL level was higher than rural students. The possible reasons were on the one hand, urban living conditions was more convenient and much the health knowledge is available, so as to help improve the city of child and adolescents HL. On the other hand, because the economy was relatively backward countryside, the popularization probability of health education and health knowledge, such as less medical treatment, education, and the educational level of students’, which results in a relatively low overall health level. However, some other survey results [29] showed that the level of HL of rural students was higher than that of urban students, possibly because of the high level of urbanization in these regions and the gradual narrowing of the gap between rural and urban health services.

The HL level of senior students was higher than that of junior students, because the more knowledge senior students received, the better they understood health knowledge and mastery of health behavior. Nevertheless, some survey results [15,16,17,18] were in conflict with this conclusion. On the one hand, it might be related to the survey methods. The survey tools of higher grades were more complex than those of lower grades. On the other hand, students in the lower grades were more flexible so that they could absorb health knowledge more easily.

Students with good academic performance had strong self-learning awareness and learning ability, and a higher degree of absorption and understanding of health knowledge, which provided a foundation for them to screen and obtain health information in daily life, and better develop behavioral habits of healthy life.

Because of their living environment and customs, Han students were more likely to acquire health knowledge.

### 4.2. Impacts of BH on HL

In terms of BH, students who paid more attention to health information [21,39,76,78] and highly exposure to online games [28,31,38] had relatively higher HL, which was consistent with relevant conclusions. Higher levels of college students’ HL were observed among those students who focused on health information, the reason was that the students with high attention to health information, for their own health and health behavior habits would also be more serious, take the initiative to expand its access to the health knowledge and source, continuously consolidate and update their health knowledge, form good life habits. Those who play online games > for 5 h per week have lower HL level than those who play online games ≤5 h. The authors suggested two possible reasons leaded to the result. On the one hand, students spent a lot of time and energy on playing games, which occupied their study time and hindered the improvement of HL. On the other hand, students played games for a long time, which indicated that their self-control ability was relatively weaker, and cannot effectively utilized health knowledge and skills.

### 4.3. Impacts of FE on HL

In terms of FE, the higher students’ HL level was observed among students that lived in a family with highly educated parents [23,25,34,40,41,43,45,51,64,65,67,69,70,71], the single child [28,40,57,76] and the higher family monthly income [27,28], which was consistent with relevant survey results. For one thing, the highly educated parents accumulated more health knowledge, which could influence and promote their children’s awareness of the importance of HL through words and deeds. And for another, parents received higher education would pay more attention to the overall cultivation of their children, and can provide better education for their children, so as to promote their children to develop a healthy lifestyle. The HL level of the single child was higher than that of not only child. In families with the single child, children occupied more family resources, so that parents would not only put efforts to cultivate their children, but also focused on the health status of their children. The level of HL increased with the increase of family monthly income. On the one hand, the improvement of economic level would improve the family’s living conditions and quality of life, and help family members to develop good habits. On the other hand, due to the development of economic level, children would gradually receive better education and learn more health knowledge.

### 4.4. Impacts of SE on HL

As far as SE is concerned, students who have received health education [22,23,25,32,60,68], majored in medicine [21,35,69,70,76,78] and went to medical school [64,74,80] are expected to have a higher level of HL, which was consistent with relevant survey results. Health education in schools is the foundation for the formation of Chinese citizens’ HL [81]. Students who have received health education courses had a higher level of HL than those who had not. Health education could significantly arouse students’ health awareness, develop good health habits, and help them choose a healthy lifestyle and cultivate behavior. Medical students’ or medical school students’ HL was comparatively higher than non-medical background students, the reasons might lay on that compared with other professional students, medical students accumulated and practiced more medical knowledge, more favorable objective conditions of health knowledge and have a more comprehensive grasp knowledge about health.

### 4.5. Study Limitations

Inevitably, this study still had some limitations. Firstly, although extensive and diverse search strategies were used to locate all possible literatures, some grey literature such as meeting minutes that are difficult to find might not be included. Secondly, the researchers found substantial heterogeneity in the estimates of Chinese students’ HL levels among the studies. This heterogeneity could due to differences in methodology between studies (including study design, research approaches and investigators) and in different educational stages (primary and secondary school students, college students). Nevertheless, it might also reflect the true state of student HL. Thirdly, since the moderating variables are not taken into account, the strength of the estimated factor may be overestimated (e.g., eastern Midwestern regions), and these factors were unlikely to be independent of each other (e.g., older urban women). Future works could provide more accurate estimates through meta-analysis of individual characteristics. Fourthly, the results of this report were that the corresponding studies were excluded through sensitivity analysis, and the reasons for the exclusion of these studies could be discussed in the future. Fifthly, some studies on factors combined ordered multiple categorical variables (for example, the overall combination of grade factors in senior three and senior two vs senior one). In the future, Corresponding data can be extracted and subgroup analysis or meta-regression analysis can be conducted on the influence degree of these factors. Finally, the Egger’s test revealed an apparent asymmetry that suggested the presence of a potential publication bias, a language bias, inflated estimates by a flawed methodologic design in smaller studies, or a lack of publication of small trials with opposite results.

## 5. Conclusions

To sum up, despite the above limitations, this research still provided some inspirations for improving the level of Chinese students’ HL and their health. First of all, the findings may help Chinese policy makers understand the overall HL of Chinese students and their levels across three dimensions (knowledge, behavior and skills). Secondly, the authors found that the IC, BH, FE and SE and student literacy level significantly correlated. In other words, students who are women, urban residents, senior class students, well performance at school, the Han nationality, focus on health knowledge, less exposure to video games, highly educated parents, income of one-child families, receiving health education, having medical background have a relatively higher HL level. Policy makers should take these factors into account when formulating relevant policies, which would help to improve the level of Chinese students’ HL, arouse students’ awareness of prevention, and lay the foundation for a healthy China.

## Figures and Tables

**Figure 1 ijerph-18-00204-f001:**
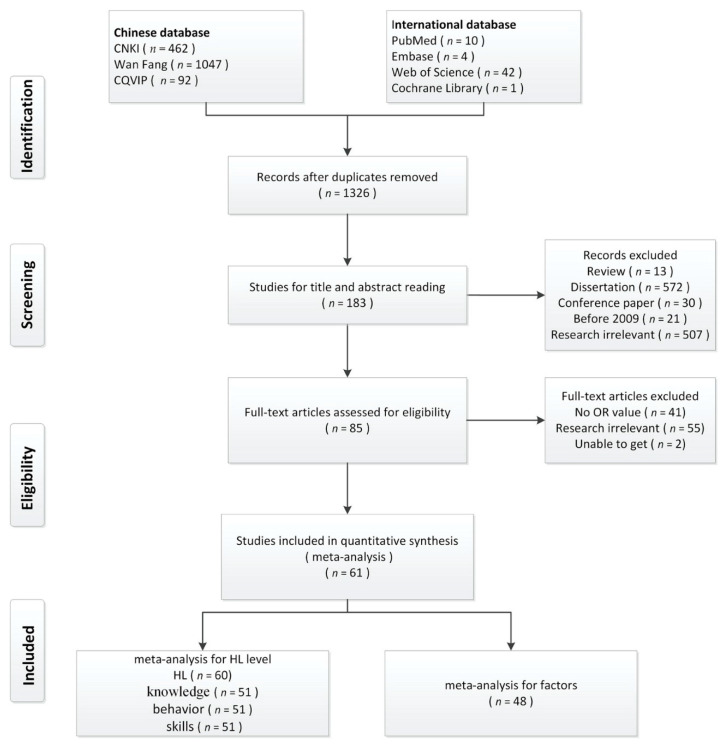
Selection process of included studies.

**Figure 2 ijerph-18-00204-f002:**
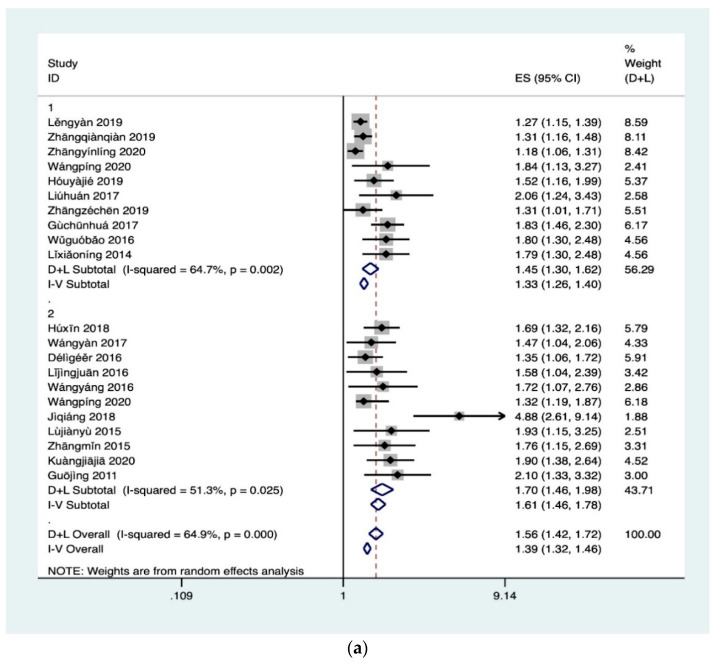
(**a**) Meta for IC. 1: gender; 2: locations. (**b**) Meta for IC. 3: grade; 4: academic performance; 5: race; (1): Sophomore year; (2): Junior year; (3): Senior year; a: excellent academic performance; b: good academic performance; c: average academic performance.

**Figure 3 ijerph-18-00204-f003:**
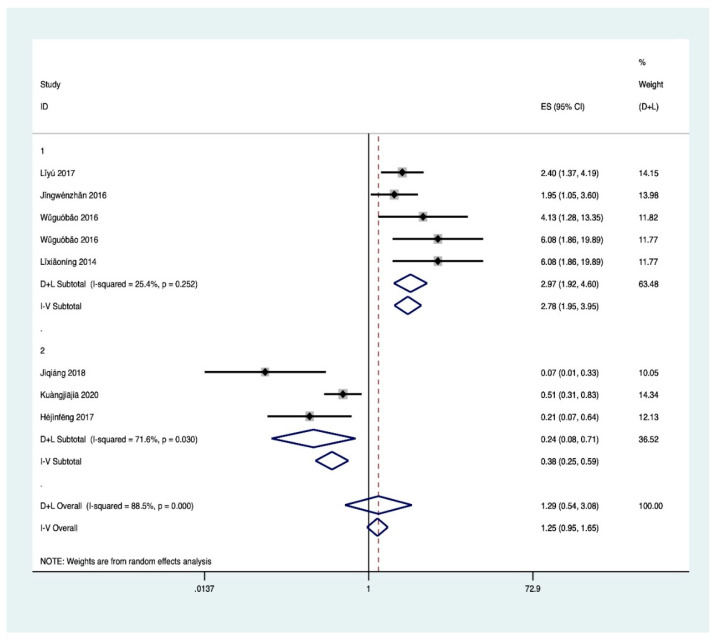
Meta for BH. 1: health information attention; 2: online game time.

**Figure 4 ijerph-18-00204-f004:**
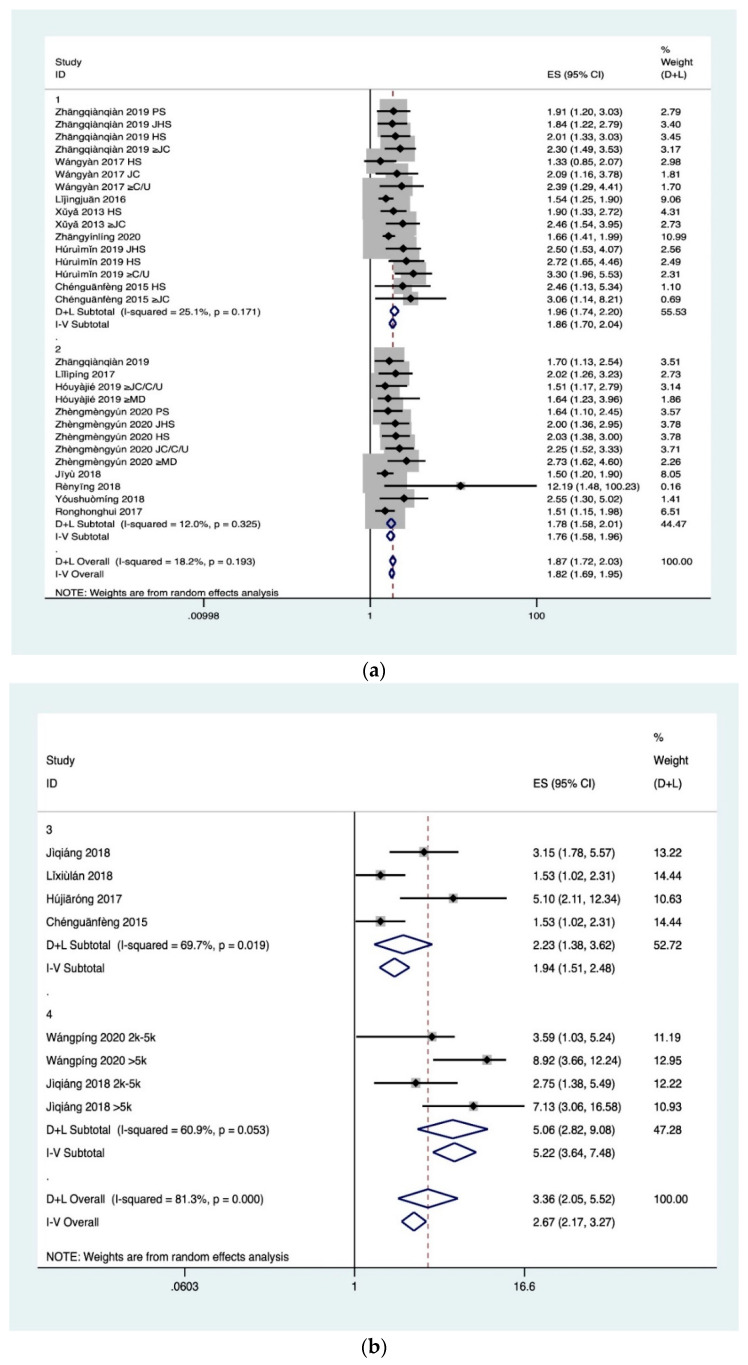
(**a**) Meta for FE. 1: father’s education level; 2: mother’s education level; HS: high school; JHS: junior high school; PS: primary school; JC: junior college; college; C: college; U: university; MD: master’s degree. (**b**) Meta for FE. 3: single-child; 4: family income; 2 k: two thousand yuan; 5 k: five thousand yuan.

**Figure 5 ijerph-18-00204-f005:**
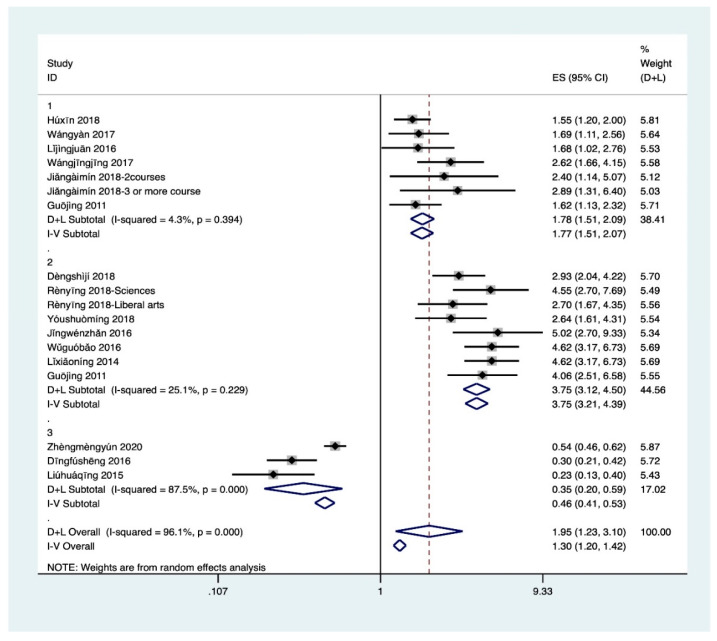
Meta for SE. 1: health education courses; 2: major; 3: school type.

**Table 1 ijerph-18-00204-t001:** Characteristics of the studies included.

Study	Publication Year	Survey Year	Region	Population	Questionnaire Recovery	Questionnaire	Sample Size	HL (%)	Knowledge (%)	Behavior (%)	Skill (%)	Meta- Anaylsis	Quality
Zhāngqiànqiàn 2018 [46]	2018	2016	Shandong	JSS	99.12	A (2015) + C + D	3588	75.45	66.69	88.82	80.21	1, 2, 3, 4	8
Lěngyàn 2019 [19]	2019	2017	Shandong	PSS	96.9	A (2015) + C	8211	59.01	52.56	75.78	55.05	1, 2, 3, 4, 5	8
Zhāngqiànqiàn 2019 [41]	2019	2016	Shandong, qingdao	PSS	99.3	A (2015) + C + D	6144	76.3	67.4	84.3	81	1, 2, 3, 4, 5	8
Húxīn 2018 [22]	2018	2016	Henan	HSS	98.12	B	1562	29.84	-	-	-	1,5	7
Lǐfèngxiá 2018 [47]	2018	2014–2015	Shandong	JSS	98.72	A + C + E	26007	13.38	12.29	15.73	50.05	1, 2, 3, 4	7
Xiāohóngxiá 2018 [48]	2018	2016	Wuhan	PSS, JSS, HSS	100	A + B (2015) + C	1300	8.23	6.15	23.12	30.31	1, 2, 3, 4	7
Pénglínlì 2017 [44]	2017	2016	Chongqing	JSS	99	F	1832	41.7	-	-	-	1	8
Wángyàn 2017 [23]	2017	2015	Guiyang	JSS, HSS	97.2	B (2014)	3295	5.5	4.6	21.1	30	1, 2, 3, 4, 5	7
Délìgéěr 2016 [24]	2016	-	Inner Mongolia	JSS	-	B	2013	19.67	17.04	22.65	77.55	1, 2, 3, 4, 5	5
Lěngyàn 2016 [49]	2016	2014–2015	Shandong	JSS	98.5	A + B (2014) + C	12458	14.6	13	17.7	50.5	1, 2, 3, 4	8
Lǐjìngjuān 2016 [25]	2016	2015	Guiyang	HSS	96.4	B (2014)	1657	6.8	4.8	26.1	36.1	1, 2, 3, 4, 5	8
Wángyáng 2016 [26]	2016	-	Dalian	PSS, JSS	-	A	405	25.7	24.4	54.6	88.1	1, 2, 3, 4, 5	6
Yángwěikāng 2015 [50]	2015	2014	Shenzhen	PSS	90.02	B (2008)	1551	10.2	5.2	28.6	70.2	1, 2, 3, 4	7
Lánlán 2014 [33]	2014	-	Sichuan, Changning	JSS, HSS	97.04	B	459	-	-	-	-	5	6
Xǔyǎ 2013 [51]	2013	2009	Guangdong	HSS	95.5	B (2008)	1606	12.3	22.4	6.9	56.4	1, 2, 3, 4, 5	7
Zhuāngrùnsēn 2013 [52]	2013	2009	Shenzhen	JSS, HSS	96.5	A	2316	24.65	40.5	20.3	84.54	1, 2, 3, 4	7
Yángruǐ 2012 [53]	2012	-	Wuhan	PSS	97.62	A (2008)	2339	39.08	62.33	68.83	27.4	1, 2, 3, 4	6
Zhāngyínlíng 2020 [43]	2020	-	Guangxi	Teenagers	95.24	F	2462	62.79	48.12	74.16	89.23	1, 2, 3, 4, 5	6
Céngqiáng 2019 [54]	2019	2016	Jiaozuo	Teenagers	95.6	F	2042	42.21	-	-	-	1	7
Húxīn 2018 [55]	2018	2017	Luoyang	JSS	93.19	F	794	38.06	-	-	-	1,5	7
Lǐlìpíng 2017 [45]	2017	2016	Shanghai	JSS, HSS	99.1	F	852	29.5	-	-	-	1,5	8
Wángpíng 2020 [27]	2020	2019	Shandong	MS	94.16	B (2015)	678	59.44	-	-	-	1,5	7
Hóuyàjié 2019 [34]	2019	2017	-	MS	97.97	A (2015)	1399	57.18	58.68	30.88	65.48	1, 2, 3, 4, 5	7
Wángyàoróng 2019 [56]	2019	-	Inner Mongolia	MS	99.52	B (2009)	420	25.95	32.14	9.76	29.76	1, 2, 3, 4	6
Dèngshìjí 2018 [35]	2018	-	Jiangsu	MS and non-MS	95.35	A	1250	20	37.1	26.7	71	1, 2, 3, 4, 5	7
Jìqiáng 2018 [28]	2018	2016	Datong	MS	92.79	B (2015)	1455	6.87	-	-	-	1,5	7
Lǐxiùlán 2018 [57]	2018	2017	Shijiazhuang	MS	91.5	B (2016)	952	14.9	43.6	12.6	31.5	1, 2, 3, 4, 5	7
Zhāngzéchēn 2018 [36]	2018	2018	Hebei	MS	-	A	1085	37.1	62.7	34.4	22.9	1, 2, 3, 4, 5	7
Zhōuwēi 2018 [58]	2018	2015	Gansu	MS	-	B (2013)	577	32.41	50.09	29.98	45.64	1, 2, 3, 4	6
Liúhuán 2017 [59]	2017	2015	Sichuan	MS	96.4	A	482	25.31	51.66	23.24	49.38	1, 2, 3, 4, 5	7
Wángjīngjīng 2017 [60]	2017	2015	Dalian	MS	91.5	B (2012)	1592	25	50.5	27.8	32.6	1, 2, 3, 4, 5	8
Péngpéng 2016 [61]	2016	-	Nanjing	MS	98.59	B (2013)	976	22.44	45.59	13.83	25.72	1, 2, 3, 4	6
Rènlìpíng 2016 [62]	2016	-	-	MS	95.6	B (2008)	956	16.53	41.21	8.26	32.22	1, 2, 3, 4	5
Lùjiànyù 2015 [29]	2015	-	Ningbo	MS	95	B (2012)	380	24.7	45	24.2	25.3	1, 2, 3, 4, 5	6
Zhāngmǐn 2015 [30]	2015	2013	Bangbu	MS and non-MS	96.61	B (2008)	1478	12.8	27.5	4.3	49.2	1, 2, 3, 4, 5	7
Huánghémèng 2014 [63]	2014	2012	Guangdong	MS	96.3	A (2008)	1260	20.23	51.03	11.43	57.94	1, 2, 3, 4	8
Zhāngzéchēn 2019 [37]	2019	-	Hebei	MS	90.4	B	1085	42.2	-	-	-	1, 5	6
Kuàngjiājiā 2020 [31]	2020	-	Hainan	CS	97.62	B (2012)	1517	11.6	25.18	23.14	16.08	1, 2, 3, 4, 5	6
Zhèngmèngyún 2020 [64]	2020	2017	Nanjing	CS	98.9	F	5644	38.1	65.2	42.6	46.9	1, 2, 3, 4, 5	7
Húruìmǐn 2019 [65]	2019	2017	Hebei	CS	89.3	B (2017)	4599	36.6	39.7	37.6	42.6	1, 2, 3, 4, 5	7
Xǔlìnà 2019 [66]	2019	2016	Beijing	CS	99.92	B (2012)	2398	44.62	56.88	46.62	43.74	1, 2, 3, 4, 5	8
Jīyù 2018 [67]	2018	-	Jiangsu	CS	96.54	B (2013)	7530	18.59	25.71	30.94	32.05	1, 2, 3, 4, 5	6
Jiǎngàimín 2018 [68]	2018	2014	Shihezi	CS	98.8	B (2008)	2057	10.7	31.1	6.1	63.6	1, 2, 3, 4, 5	8
Rènyīng 2018 [69]	2018	2017	Hengyang	CS	97.23	B (2009)	1055	13.08	20.66	3.03	78.96	1, 2, 3, 4, 5	7
Yóushuòmíng 2018 [70]	2018	2017	Shanghai	CS	99.48	B (2008)	761	27.7	39.7	4.9	74.4	1, 2, 3, 4, 5	8
Ronghonghui 2017 [71]	2017	-	Chongqing	CS	93.34	B (2012)	3183	21.05	34.6	26.36	22.62	1, 2, 3, 4, 5	7
Dùguópíng 2017 [72]	2017	2015	Jiangsu	CS	94.7	B (2009) + C	7560	40.4	56.9	34.2	76.1	1, 2, 3, 4, 5	8
Gùchūnhuá 2017 [20]	2017	2016	Shanghai	CS	-	B	1405	2.35	-	-	-	1, 5	6
Héjìnfēng 2017 [38]	2017	2017	Jiangsu	CS	91.86	B (2012)	1027	13.24	15.77	15	7.3	1, 2, 3, 4, 5	7
Lǐyú 2017 [39]	2017	2015	Shanghai	CS	95. 2	B (2014) Beijing	457	16. 19	36.76	11.82	16.85	1, 2, 3, 4, 5	7
Rènhuānhuān 2017 [73]	2017	-	Lanzhou	CS	86.18	A + B (2015)	1571	7.1	6.7	35.5	7.8	1, 2, 3, 4	6
Dīngfúshēng 2016 [74]	2016	-	Wuhu	CS	98.1	B (2009)	706	29	38.8	6.5	75.1	1, 2, 3, 4, 5	7
Hújiāróng 2016 [75]	2016	-	-	CS	96.4	A + B (2015)	484	15.91	43.39	11.16	83.47	1, 2, 3, 4	6
Jǐngwénzhǎn 2016 [76]	2016	2014	Nanjing, Chengdu, Guangzhou, Beijing, Xi’an	CS	95.7	B (2008)	850	15.3	41.9	10	55.9	1, 2, 3, 4, 5	7
Rènlìpíng 2016 [77]	2016	2015	Jilin	CS	97.81	A (2008)	1700	10.82	10	18.53	53.94	1, 2, 3, 4	7
Wǔguóbǎo 2016 [78]	2016	-	Xinjiang Uygur	CS	97.93	A (2008)	1942	9.22	20.8	5.2	68.95	1, 2, 3, 4, 5	6
Chénguānfèng 2015 [40]	2015	2014	Ganzhou	CS	94.3	B (2008)	1132	5.21	21.91	7.24	26.77	1, 2, 3, 4, 5	7
Dáyīngjuān 2015 [79]	2015	2014	Shanghai	CS	98.02	B (2008)	2084	66.65	50.67	65.6	74.38	1, 2, 3, 4	7
Liúhuáqīng 2015 [80]	2015	-	Bangbu	CS	96.61	B (2008)	2478	8.9	19.8	3.3	46.9	1, 2, 3, 4, 5	6
Lǐxiǎoníng 2014 [21]	2014	2010	Guangzhou	CS	98.2	B (2008)	2150	8.84	22.6	5.7	70.5	1, 2, 3, 4, 5	7
Guōjìng 2011 [32]	2011	-	Beijing	CS	90.5	B (2009)	905	24.75	41.77	13.37	73.04	1, 2, 3, 4, 5	6

PSS: Primary school students; JSS: Junior school students; HSS: High school students; MS: Medical students; CS: College students;1: HL; 2: knowledge; 3: behavior; 4: skills; 5:factors; A: Chinese citizens’ health literacy—basic knowledge and skills; B: Chinese citizen health literacy questionnaire; C: Guidelines for health education in primary and secondary schools; D: Norms of health education in primary and secondary schools; E: Guidelines for mental health education in primary and secondary schools; F: Self-made questionnaire.

## Data Availability

No new data were created or analyzed in this study. Data sharing is not applicable to this article.

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
