# Peer review of "Chinese Students’ Health Literacy Level and Its Associated Factors: A Meta-Analysis"

_ijerph, 2020, doi:10.3390/ijerph18010204_

Round 1

Reviewer 1 Report

General

  1. Please remove systematic review from the title as this is a meta-analysis.
  2. To be consistent, please indicate a p-value as a letter but not capital.
  3. You may replace “influencing factor” with “associated factors” as we can not infer causality via observational studies.  

Abstract

  1. Line 10: Please spell out HL as it came to the abstract at the first time.

Introduction

  1. Compared to 2018, in 2019 the basic knowledge and ideological level of Chinese residents was 34.31%, which saw an increment of 3.79 percentage points; The percentage of the level of healthy lifestyle and behavioral literacy increased by 2.44 percentage points at 19.48%; The level of basic skills and literacy was 21.43%, which experienced a growth of 2.75 percentage points.

  • Compared to 2018, in 2019 the basic knowledge and ideological level of Chinese residents was 34.31%, which saw an increment of 3.79 percentage points. The percentage of the level of healthy lifestyle and behavioral literacy increased by 2.44 percentage points at 19.48%. The level of basic skills and literacy was 21.43%, which experienced a growth of 2.75 percentage points.
  • Please provide appropriate references to support the argument.

Methods

  1. Line 99: Based on the guidelines19 for systematic evaluation and meta-analysis…

→ Based on the guidelines for systematic evaluation and meta-analysis…

  1. Lin 112-119: Please revise letters and capitals per grammar. Please spell out E-HL.
  2. Line 132-140: Please cite appropriate references corresponding to their statements.

Results

  1. Line 159: 95 full-text articles were…

→ Eighty-five full-text articles were…

  1. How did you calculate the rate of HL, knowledge, behavior, and skills across the studies using different instruments?
  2. Line 186: Please replace “p = 0.000” with “p < 0.001.”

Discussion

  1. What is the uniqueness generated from your study? What are your suggestions regarding implementations for clinical practice and future study? You have to discuss more information relevant to your research question and provide concrete suggestions to enrich existing knowledge of patient care.

Figure

  1. Figure 1: The number of full-text articles assessed for eligibility should be 85 but not 95.

Reference

Reference styles of #1, 2, 5, 10, 17, 18, 22, 31, 32, 42, are not aligned with the journal guide; please correct it per journal requirement.

Author Response

Response to Reviewer 1 Comments

Re: Re-submission of Manuscript ID ijerph-1033988

On behalf of my research team, I would like to thank you for taking your time and efforts in reviewing our manuscript. We have revised our manuscript in accordance with your comments and suggestions, and please find below our point by point responses. We hope that you will find our work suitable for publication in IJERPH.

Point 1: General-1. Please remove systematic review from the title as this is a meta-analysis.

Response 1: We have removed systematic review

Point 2: General- 2. To be consistent, please indicate a p-value as a letter but not capital.

Response 2: We have consistently used the lowercase p to indicate p-value.

Point 3: General- 3. You may replace “influencing factor” with “associated factors” as we can not infer causality via observational studies.

Response 3: We have replaced “influencing factor” with “associated factors”

Point 4: Abstract- 1. Line 10: Please spell out HL as it came to the abstract at the first time.

Response 4: We have spelled out health literacy in the abstract.

Point 5: Introduction-1. Compared to 2018, in 2019 the basic knowledge and ideological level of Chinese residents was 34.31%, which saw an increment of 3.79 percentage points; The percentage of the level of healthy lifestyle and behavioral literacy increased by 2.44 percentage points at 19.48%; The level of basic skills and literacy was 21.43%, which experienced a growth of 2.75 percentage points.

Please provide appropriate references to support the argument.

Response 5: We have provided appropriate references to support the argument.

Publicity Department of The National Health Commission. The national health literacy level rose to 19.17 percent in 2019. http://www.nhc.gov.cn/xcs/s3582/202004/df8d7c746e664ad783d1c1cf5ce849d5.shtml (accessed on 10 Nov 2020)

Point 6: Methods- 1. Line 99: Based on the guidelines19 for systematic evaluation and meta-analysis…

→ Based on the guidelines for systematic evaluation and meta-analysis…

Response 6: We have revised it according to the comments.

Point 7: Methods- 2. Lin 112-119: Please revise letters and capitals per grammar. Please spell out E-HL.

Response 7: We have revised letters and capitals per grammar. As suggested, we have spelled out e-HL in the paper.

Point 8: Methods- 3. Line 132-140: Please cite appropriate references corresponding to their statements.

Response 8: After much deliberation, we found this description inappropriate and therefore deleted this statement. In addition, AHRQ references are cited.

Point 9: Results-1. Line 159: 95 full-text articles were…

→ Eighty-five full-text articles were…

Response 9: It has been revised according to the comments.

Point 10: Results-2. How did you calculate the rate of HL, knowledge, behavior, and skills across the studies using different instruments?

Response 10:First, we combined the data (HL, knowledge, behavior, and skills have been given in each study) in 61 papers through meta analysis to calculate the final rate of HL, knowledge, behavior, and skills. No forest map is provided in the paper due to space limitations. The forest map is as follows:(submit). Secondly, although the tools of different studies are not completely consistent, they are all based on the basic Knowledge and Skills of Chinese Citizens' Health Literacy as the initial basis and combined with the Questionnaire on Chinese Citizens' Health Literacy.

Point 11: Results-3. Line 186: Please replace “p = 0.000” with “p < 0.001.”

Response 11: It has been revised according to the comments.

Point 12: What is the uniqueness generated from your study? What are your suggestions regarding implementations for clinical practice and future study? You have to discuss more information relevant to your research question and provide concrete suggestions to enrich existing knowledge of patient care.

Response 12: As mentioned at the beginning of the Discussion section, our study is the first to meta-analysis on the associated factors of Chinese students' HL. Our findings can contribute to devising strategies, in particular about the study from this perspective provides scientific basis for targeted intervention of Chinese students' HL. In addition , the meta-analysis to study the potential relationship between these variables and HL can provide a valuable reference for the study of whether they are relevant and whether they should be included in the associated factors of HL.

Point 13: Figure 1: The number of full-text articles assessed for eligibility should be 85 but not 95.

Response 13: It has been revised according to the comments.

Point 14: Reference styles of #1, 2, 5, 10, 17, 18, 22, 31, 32, 42, are not aligned with the journal guide; please correct it per journal requirement.

Response 14: It has been revised according to the journal requirement.

Reviewer 2 Report

The text has many qualities of a professional review, but in this form it is difficult to review it. The description is not clear enough. As can be seen from the diagram, 61 relevant papers have been identified, which are shown in the table. The references are not consistent with this list. Apart from the name of the author and the year of publication, there should be a reference to the literature as [ ]. The selection has been  intended for the Chinese population, and both  English and Chinese  papers are acceptable. If a reviewer cannot reach these papers or does not speak Chinese, it is difficult to give an opinion. Please specify how many papers were in English. A list of 61 papers could even be included in the annex, in addition to the main reference list, with translated tile and information of language. Please link the items fro this annex with table 1. 

The lack of definition of the basic concepts and the lack of methodological consistency are striking. HL is expressed as %. Is it % of the maximum score for any one questionnaire? In other words, the index is standardized on a scale of 0-100. In the table with 61 papers, the name of the questionnaire is not provided.  The paper is intended to address the determinants, and the main result is the HL level itself resulting from the meta-analysis. The availability of data on the OR is given as the inclusion criterion. Meanwhile, in the figures with metha-analysis is ES, as an unexplained abbreviation (effect size). The figures with meta  analysis cover much less than 61 works. On the bottom of flowchart the number of  papers with aby information on determinants is expected.  In the title of the figures, I suggest giving the number of papers  being the basis of the meta-analysis.  

There is a reference to STORBE as a standard for quality assessment of observational studies. There should also be a reference to PRISMA, because the flowchart has a recommended form

The division into IC, BH, FE, SE factors should be introduced in the methods. In the table with abbreviations FB should be changed to FE.

The figures show unexplained abbreviations.

The review was planned to cover the studies published between 2010 and 2020. in fact, most of the publications are from recent years. It is worth emphasising this. 

Author Response

Response to Reviewer 2 Comments

Re: Re-submission of Manuscript ID ijerph-1033988

On behalf of my research team, I would like to thank you for taking your time and efforts in reviewing our manuscript. We have revised our manuscript in accordance with your comments and suggestions, and please find below our point by point responses. We hope that you will find our work suitable for publication in IJERPH.

Point 1: The text has many qualities of a professional review, but in this form it is difficult to review it. The description is not clear enough. As can be seen from the diagram, 61 relevant papers have been identified, which are shown in the table. The references are not consistent with this list. Apart from the name of the author and the year of publication, there should be a reference to the literature as [ ]. The selection has been intended for the Chinese population, and both English and Chinese  papers are acceptable. If a reviewer cannot reach these papers or does not speak Chinese, it is difficult to give an opinion. Please specify how many papers were in English. A list of 61 papers could even be included in the annex, in addition to the main reference list, with translated tile and information of language. Please link the items fro this annex with table 1.

Response 1:We added references to each study in table1. We searched a lot of literature. The search results of English literature were shown in Figure 1, and only one English literature was included in the end. We provide the corresponding Chinese literature and the corresponding translated English literature according to table1

Point 2: The lack of definition of the basic concepts and the lack of methodological consistency are striking. HL is expressed as %. Is it % of the maximum score for any one questionnaire? In other words, the index is standardized on a scale of 0-100. In the table with 61 papers, the name of the questionnaire is not provided.  The paper is intended to address the determinants, and the main result is the HL level itself resulting from the meta-analysis. The availability of data on the OR is given as the inclusion criterion. Meanwhile, in the figures with metha-analysis is ES, as an unexplained abbreviation (effect size). The figures with meta  analysis cover much less than 61 works. On the bottom of flowchart the number of  papers with aby information on determinants is expected.  In the title of the figures, I suggest giving the number of papers  being the basis of the meta-analysis. 

Response 2: First of all, the standard of health literacy for a sample in most studies is to correctly answer 80% or more of the health literacy questions. Secondly, the ratio of the sum of these samples with health literacy to the total samples is the health literacy rate. The questionnaires of all the studies are based on the basic Knowledge and Skills of Chinese Citizens' Health literacy as the initial basis and formulated in combination with the Questionnaire on Chinese Citizens' Health Literacy. Therefore, we do not list the names of questionnaires in table1.In the selection criteria, we have set the OR value of factors as one of the inclusion criteria, while HL is the combination of rates and does not involve the OR value. The specific forest map is shown in the figure below(submit). All 61 references were used for meta-analysis. See the updated Figure 1 and Table 1 for the specific structure.Based on your suggestions, the meanings of the unexplainable symbols in each diagram have been given.

Point 3: There is a reference to STORBE as a standard for quality assessment of observational studies. There should also be a reference to PRISMA, because the flowchart has a recommended form

Response 3: After much deliberation, we found this description inappropriate , thus we deleted this statement.

Point 4: The division into IC, BH, FE, SE factors should be introduced in the methods. In the table with abbreviations FB should be changed to FE.

Response 4: It has been revised according to the comments.

Point 5: The figures show unexplained abbreviations.

Response 5: It has been revised according to the comments.

Point 6: The review was planned to cover the studies published between 2010 and 2020. in fact, most of the publications are from recent years. It is worth emphasising this.

Response 6: The time frame we searched for was indeed between 2010 and 2020, but most of the studies we screened for eligibility were in the last few years. According to your comments, we have specified this point in section ‘3.1. Characteristics of the Study Samples’.

Reviewer 3 Report

very interesting and well-written paper. I only corrected few mistakes in the attached file

Author Response

Re: Re-submission of Manuscript ID ijerph-1033988

On behalf of my research team, I would like to thank you for taking your time and efforts in reviewing our manuscript. We have revised our manuscript in accordance with your comments and suggestions, and please find below our point by point responses. We hope that you will find our work suitable for publication in IJERPH.

Point 1: Very interesting and well-written paper. I only corrected few mistakes in the attached file

Response 1: We have modified it according to your suggestion

Round 2

Reviewer 1 Report

Line 34: At present, experts and scholars have gradually reached a consensus on the definition of HL.

→ At present, experts and scholars have gradually reached a consensus on the definition of health literacy (HL).

Author Response

Re: Re-submission of Manuscript ID ijerph-1033988

On behalf of my research team, I would like to thank you for taking your time and efforts in reviewing our manuscript. We have revised our manuscript in accordance with your comments and suggestions, and please find below our point by point responses. We hope that you will find our work suitable for publication in IJERPH.

Point 1: Line 34: At present, experts and scholars have gradually reached a consensus on the definition of HL.

→ At present, experts and scholars have gradually reached a consensus on the definition of health literacy (HL).

Response 1: It has been revised according to the comments.

Reviewer 2 Report

The article is now a more reliable review, because the list of papers qualified for meta-analysis seem to be complete. There are, however, still many issues to be clarified, and many editorial shortcomings.

Still not explained what HL% means in Table 1 (eighth column). As it is the main outcome from every reported study it should be explained in methods. Without explanation the main results is not clear (The results showed that the level rates of HL and its three dimensions were 26% (95% CI,21%-30%), 35% (95% 21 CI,29%-40%), 26% (95% CI,19%-33%), 51% (95% CI,45%-57%), respectively). Is it really level rate of HL?

HL research is based on questionnaires. I do not see a list of national tools with their short description, which were used in individual studies.

The figures show ES in the meta-analysis and the methods show OR (row 150).

OR and ES (see figures) not explained in abbreviations. OR never used after announcing in methods section.

The abstract should be much shorter according to instruction - of about 200 words maximum. A 10-15% margin can be accepted, but in this version it has about 300 words.

Reference [4] seems to be incomplete.

Line 133 is unreadable

There are many editing errors in that there are no spaces between the words (e.g. line 141)

The numbering of the figure is unclear, two have number 2 and two have number 4.

Author Response

Re: Re-submission of Manuscript ID ijerph-1033988

On behalf of my research team, I would like to thank you for taking your time and efforts in reviewing our manuscript. We have revised our manuscript in accordance with your comments and suggestions, and please find below our point by point responses. We hope that you will find our work suitable for publication in IJERPH.

Point 1: Still not explained what HL% means in Table 1 (eighth column). As it is the main outcome from every reported study it should be explained in methods. Without explanation the main results is not clear (The results showed that the level rates of HL and its three dimensions were 26% (95% CI,21%-30%), 35% (95% 21 CI,29%-40%), 26% (95% CI,19%-33%), 51% (95% CI,45%-57%), respectively). Is it really level rate of HL?

Response 1: Health literacy % means the prevalence rate of health literacy, which is the level rate of health literacy (Chinese paradigm). The level rate of HL is the result of combining a number of studies, so we have reason to believe that this is indeed the level rate of HL

Point 2: HL research is based on questionnaires. I do not see a list of national tools with their short description, which were used in individual studies.

Response 2: It has been revised in Table 1 (seventh column)

Point 3: The figures show ES in the meta-analysis and the methods show OR (row 150).

OR and ES (see figures) not explained in abbreviations. OR never used after announcing in methods section.

Response 3:

(1) ES = effect size, OR = odds ratio. It could seen in abbreviations.

(2) ES here refers to the OR.  I've explained this on row 202.

Point 4: The abstract should be much shorter according to instruction - of about 200 words maximum. A 10-15% margin can be accepted, but in this version it has about 300 words.

Response 4: The abstract has been reduced to about 270 words

Point 5: Reference [4] seems to be incomplete.

Response 5: It has been revised.

Point 6: Line 133 is unreadable

Response 6: It has been revised.

Point 7: There are many editing errors in that there are no spaces between the words (e.g. line 141)

Response 7: It has been revised.

Point 8: The numbering of the figure is unclear, two have number 2 and two have number 4.

Response 8: It has been revised.
